# Comprehensive Review on Research Status and Progress in Precision Grinding and Machining of BK7 Glasses

**DOI:** 10.3390/mi15081021

**Published:** 2024-08-09

**Authors:** Dayong Yang, Zhiyang Zhang, Furui Wei, Shuping Li, Min Liu, Yuwei Lu

**Affiliations:** School of Mechanical and Automotive Engineering, Guangxi University of Science and Technology, Liuzhou 545006, China; dyyang@gxust.edu.cn (D.Y.); 20230102050@stdmail.gxust.edu.cn (Z.Z.); 100001550@gxust.edu.cn (S.L.); vera_lu@live.cn (Y.L.)

**Keywords:** BK7 glass, hard and brittle materials, precision grinding, precision machining, surface quality, ultrasound-assisted machining

## Abstract

BK7 glass, with its outstanding mechanical strength and optical performance, plays a crucial role in many cutting-edge technological fields and has become an indispensable and important material. These fields have extremely high requirements for the surface quality of BK7 glass, and any small defects or losses may affect its optical performance and stability. However, as a hard and brittle material, the processing of BK7 glass is extremely challenging, requiring precise control of machining parameters to avoid material fracture or excessive defects. Therefore, how to obtain the required surface quality with lower cost machining techniques has always been the focus of researchers. This article introduces the properties, application background, machining methods, material removal mechanism, and surface and subsurface damage of optical glass BK7 material. Finally, scientific predictions and prospects are made for future development trends and directions for improvement of BK7 glass machining.

## 1. Introduction

The development of BK7 glass originated from the advancement of optical glass technology. In the early days, it was discovered that by adding different oxides to glass, its refractive index and dispersion performance could be changed, thereby making the resulting manufactured glass suitable for various optical systems. The discovery and application of this technology laid the foundation for the birth of BK7 glass. BK7 glass has developed in this context, playing a crucial role in the field of optics due to its unique chemical composition, excellent physical properties, and optical properties. In 1965, Japan’s HOYA company was the first to achieve continuous melting production of BK7 optical glass without stripes or bubbles. The successful application of this technology has greatly improved the quality of BK7 glass and reduced production costs. BK7 glass, a high-performance optical glass, has held an important position in the field of optics since its inception.

As an excellent optical element, the bubble-free characteristics of BK7 glass ensure the pure transmission of light, avoiding scattering and loss caused by bubbles [1]. High linear transmittance means that BK7 glass can maximize the transmission of light, reduce the loss of light energy, and improve the overall efficiency of the optical system [2]. At the same time, its high refractive index makes BK7 glass more flexible and accurate in refracting and reflecting light, further improving the performance of optical systems [3]. In addition to its excellent optical properties, BK7 glass also exhibits a series of remarkable physical and chemical properties. First, its conductivity is extremely low and almost negligible, which renders BK7 glass an excellent insulator for applications requiring such properties. Second, BK7 glass exhibits excellent corrosion resistance [4] and can maintain its stability for a long time in various corrosive environments without being corroded by chemical substances. Third, BK7 glass also exhibits good wear resistance [5], maintaining its surface smoothness and clarity even under frequent friction and wear. These characteristics enable BK7 glass to maintain stable performance in harsh environments and extend its service life. Because of these characteristics, BK7 glass has been processed into optical lenses for application in aerospace systems [2], astronomical observation systems, laser fusion devices, biomedical imaging [6,7], precision optical measurement instruments [8], and other fields. To maximize the advantages of BK7 glass in various fields, it is necessary to further machine BK7 glass to produce high-quality surface parts with a certain dimensional accuracy and geometric shape accuracy, producing a high-precision surface with minimal defects. These high-precision surfaces can further reduce light scattering and loss and improve the performance of optical systems. At present, grinding, polishing, laser machining [9], electrochemical discharge machining [10,11], and other machining methods are usually used for the machining of optical glass. In order to improve the surface quality of workpieces, scholars have improved and optimized the above machining methods, e.g., with microgrinding [12], ultrasound-assisted grinding, ultrasound-assisted polishing [13], picosecond laser-assisted polishing [14], and in situ laser-assisted diamond cutting [15].

Therefore, to improve machining efficiency, reduce the scrap rate, and improve product quality, it is necessary to have a deeper understanding of the processing mechanisms for BK7 glass. This article summarizes the research status and progress of BK7 glass in recent years and suggests directions for future research and development.

## 2. The Application and Common Machining Methods for BK7

The full name of BK7 glass is borosilicate crown glass. Figure 1a shows the macroscopic structure of BK7 glass, and Figure 1b shows the surface morphology after polishing. As shown in the energy spectrum of BK7 glass in Figure 1c, it can be inferred that BK7 glass is mainly composed of boron, silicon, aluminum, sodium, calcium, and other components, with an atomic structure as shown in Figure 1d. These components and structures endow it with unique optical and physical properties. The main oxides introduced into BK7 glass include silicon dioxide (SiO_2_) [16], boron trioxide (B_2_O_2_), aluminum oxide (Al_2_O_2_), sodium oxide (Na_2_O), and calcium oxide (CaO) [17]. The introduction of these oxides not only changes the chemical composition of glass but also has a profound impact on its physical and optical properties. Specifically, silicon dioxide is the main component of glass, which endows it with basic mechanical strength and chemical stability. The addition of B_2_O_2_ increases the high-temperature resistance and chemical stability of glass, enabling it to maintain a stable performance in high-temperature environments. The introduction of Al_2_O_2_ regulates the refractive index and scattered light properties of the glass, enabling BK7 glass to achieve high-precision light focusing and imaging in optical systems. The addition of Na_2_O and CaO improves the thermal expansion coefficient and optical properties of the glass, giving BK7 glass a lower thermal expansion coefficient and good optical uniformity.

BK7 has been applied in multiple fields. In the field of nuclear energy and nuclear engineering, the world’s largest laser confinement fusion device (National Ignition Facility (NIF) equipment) used 1600 mirrors and polarizers made of BK7 optical glass material [20]. In the field of lasers [2], BK7 glass can be used in key components such as output windows and laser prisms [21], which can effectively reduce energy loss during laser transmission while maintaining the stability and directionality of the laser beam. In the aerospace industry [2], BK7 glass is used to make various optical components [4], such as lenses [22,23]. These components can maintain stable optical performance even under extreme environmental conditions, such as high temperature, low temperature, and strong radiation. In the field of medicine [24], BK7 glass can be used to make lenses and optical sensors [25,26] for endoscopes in medical detection equipment. In the field of communication, BK7 glass can be used as a material for making optical fibers [27]. In photography equipment, lenses or optical components are usually used as equipment such as photography lenses, filters, and viewfinders. Its excellent performance provides photographers with high-quality, realistic, and artistic photos. Figure 2a shows the camera lens of BK7 glass. In the manufacturing of microscopes, it can be used as a lens material for objective lenses, eyepieces, and other optical components, providing high-quality imaging effects for scientific research and medical fields. Figure 2b illustrates the utilization of BK7 glass in the field of optical microscopes. In the field of astronomy, BK7 glass can effectively reduce light energy loss, improve the transmission efficiency of optical systems, and enable astronomers to obtain clear and accurate star images. This makes it an ideal material for manufacturing telescope lenses, ensuring that astronomers can obtain clear and accurate star images. Figure 2c shows the primary mirror of the BK7 glass space telescope. In terms of solar panels, BK7 glass has excellent transparency, which can effectively convert sunlight into electricity, thereby improving the conversion efficiency of solar panels and better absorbing and utilizing solar energy. Secondly, it also has good mechanical properties and can withstand certain pressures and impacts. This makes the application of BK7 glass in solar panels safer and more reliable, able to withstand the effects of harsh environments and weather conditions, as shown in Figure 2d of the solar panel made of BK7 glass.

The high hardness, significant brittleness characteristics, and relatively low fracture toughness of BK7 glass pose significant challenges during its processing [28]. At present, several traditional machining techniques, such as cutting [29], milling [30,31], and drilling techniques [32,33,34], as well as many nontraditional machining techniques, such as laser machining [35,36], abrasive flow machining (AFM) [37], discharge machining [38,39], 3D printing [40,41], wire cutting machining [42], chemical mechanical polishing, and chemical etching machining [43], are widely used in the machining of the surface microstructure of BK7 glass. As shown in Figure 3, there are four common machining methods: grinding, ultrasound-assisted machining, laser machining, and wire-cutting machining. However, this inherently fragile material has problems with long production cycles and has low economy during machining [44]. The machining of BK7 glass requires strict control of the amount of material removed from the surface of the workpiece to finely adjust its surface quality and achieve the high precision required. In the process of manufacturing high-precision glass optical components, grinding technology is often regarded as a core and critical machining method. This approach can not only ensure that the surface quality of optical components reaches extremely high standards but also effectively improve their optical performance [45,46], making it an excellent machining method [47].

To accurately control the machining parameters, optimize the processing flow, and ensure the high precision and quality of BK7 glass optical components during processing, it is necessary to have a deeper understanding of the machining mechanism for BK7 glass during the grinding process. However, material removal during the grinding process is a complex and variable process, reflected in the randomly distributed scratches formed by the interaction between the workpiece surface and multiple sand particles and gravel. It is very difficult to directly study the complex interaction mechanism between gravel and workpieces. Usually, scholars simulate the contact between gravel and a workpiece surface during an actual grinding process through scratch experiments and directly observe and analyze various phenomena under this interaction to infer complex grinding processes [28,48].

**Figure 3 micromachines-15-01021-f003:**
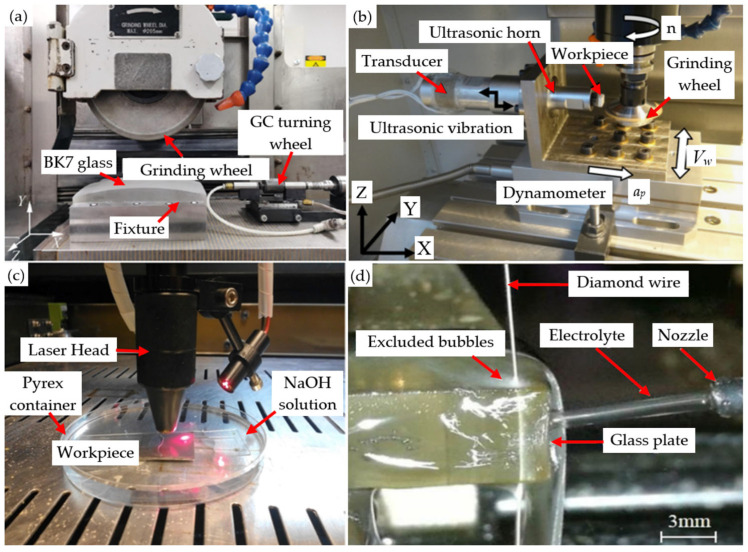
Four common machining methods: (**a**) grinding machining [3]; (**b**) ultrasound-assisted machining [49]; (**c**) laser machining [50]; (**d**) wire cutting [51].

## 3. Mechanism of BK7 Glass Material Removal

The material removal mechanism occupies a core position in the field of mechanical processing, and it is the theoretical cornerstone for understanding and analyzing the machining process. Understanding the removal mechanism of BK7 glass can optimize its machining process. By conducting in-depth research on the behavior of materials during the machining process, the optimal machining parameters, such as cutting speed, cutting depth, and tool selection, can be attained to ensure machining efficiency, improve machining quality, and reduce possible defects or damage during the machining process. By studying the removal mechanism of BK7 glass, we can better understand the possible errors and defects that may occur during machining, find ways to reduce or eliminate these errors and defects, improve machining quality, and explore new machining technologies and methods.

### 3.1. Common Scratch Devices and BK7 Scratch Experiments

The removal of surface material from BK7 glass can be achieved through grinding [47], thereby obtaining work parts with high surface quality, a smooth surface, and high shape accuracy [44]. Scratch experiments play a role in revealing the mechanism of grinding processes. By simulating the actual contact and interaction between abrasive particles and workpieces, scratch experiments not only provide us with an effective means to verify the interaction mechanism between the two during the grinding process but also further elucidate the complex behavior of material removal from BK7 glass during grinding [52,53], making them an effective experimental research method [22,54]. The material scratch test is an important testing method used to determine the surface hardness, wear resistance, and scratch resistance of materials. Figure 4a shows a schematic diagram of a scratch experiment. Figure 4b–d are common scratch experimental devices, which are described as follows: nanoscratch device, single abrasive scratch device, and ultrasonic scratch device.

The basic principle of this experimental method is to apply a certain amount of pressure and use a scratch tool with a certain shape to scratch the surface of the material. Subsequently, by observing and measuring the length, depth, and shape of the scratches, the hardness, wear resistance, and other key mechanical properties of the material can be analyzed. As an important tool for analyzing the surface properties of materials, material scratch detection has been applied in multiple fields, especially in materials science, engineering, and mechanical manufacturing. It can be used not only to evaluate the hardness, wear resistance, and scratch resistance of various solid materials but also to study the deformation behavior, fracture mechanism, and surface treatment effect of materials. In the fields of materials science, mechanical engineering, and the automotive industry, material scratch experiments play an important role, providing strong technical support for material selection, machining optimization, and product improvement.

#### 3.1.1. Single-Scratch Experiment on BK7 Glass

The unique high hardness and relatively low fracture toughness of BK7 glass pose a significant challenge during precision machining. These physical properties require both sufficient precision to effectively remove materials during processing and advanced technology to avoid excessive cracks and microcracks. Therefore, the accuracy and progressiveness of processing technology are crucial for ensuring the processing quality of BK7 glass [56]. Studying the crack propagation and material removal mechanism of optical glass through scratch experiments is highly important for improving machining accuracy. Scratches on hard and brittle materials are not simple machining, but are influenced by various factors (such as scratch speed, scratch direction, tool geometry and parameters, and environment). The observation indicators used in scratch experiments mainly include scratch depth, scratch morphology, and crack formation. The cracks generated by the scratch experiment include lateral cracks, radial cracks, and median cracks. Lateral and radial cracks can be observed in Figure 5a,b. In Figure 5c, both radial and median cracks can be observed. Finally, in Figure 5d, lateral cracks, radial cracks, and median cracks can be observed simultaneously.

Usually, the scratch process under increasing load includes several significant stages. At extremely low loads, the material mainly undergoes elastic deformation, which means that it can be restored to its original state without permanent deformation. However, as the load gradually increases, the material begins to exhibit plastic deformation, which is permanent, but no cracks appear at this time. Subsequently, when the load reaches a critical value, that is, after experiencing the transition point from toughness to brittleness, intermediate cracks begin to appear in the scratches, which is a clear sign of brittle fracture in the material. As the load continues to increase, transverse cracks begin to form around the scratch, and then these cracks propagate to the surrounding areas, forming radial cracks. The propagation of these cracks further intensifies the failure of the material [60]. The removal of material from the glass during precision machining can be broadly classified into two categories: ductile and brittle modes. These modes exhibit distinct characteristics and effects during the machining process. In the toughness mode, the removal process of glass materials is influenced by various factors, among which elastic recovery and plastic uplift are the two main key factors. In the brittle mode, material delamination (lateral crack peeling and radial crack peeling) is the main form of removal [59]. During the nanoscratch machining process of BK7 glass, there may be differences between the actual depth of the scratch and the theoretical depth. This phenomenon is due to the elastic recovery of the material during the scratch process, which directly affects the machining accuracy [61]. Wang et al. [57,58] thoroughly considered the elastic recovery rate and friction characteristics of BK7 glass during nanoscratch processing and conducted experimental analysis on two different scratch directions (face-forward and edge-forward). Research has shown that the elastic recovery rate and residual stress of face-forward scratched materials are significantly greater. This means forward scratches are more susceptible to transverse cracks and further propagation than lateral scratches, ultimately leading to greater material removal.

#### 3.1.2. Multiple-Scratch Experiments on BK7 Glass

Through detailed research on single-scratch tests, the complex mechanism of BK7 glass material removal has been preliminarily revealed. However, there is a strong interaction between the sand and gravel of the grinding wheel, resulting in different patterns of crack initiation and propagation [62,63], which is difficult to reflect in single-scratch experiments. To reveal the interaction between closely arranged abrasive particles, many scholars have conducted studies on double and continuous scratches. Through these studies, a deeper understanding of the processing behavior of BK7 glass has been gained, and important theoretical support has been provided for optimizing the processing parameters, improving the processing quality, and extending the tool life [64]. From the perspective of the experimental results and data analysis, double-scratch and continuous-scratch experiments can provide more information about material properties. By comparing the results of the scratches, we can gain a deeper understanding of the hardness, wear resistance, fracture toughness, and other properties of the material. Compared to a single scratch, multiple scratches will generate greater maximum principal stresses during the machining process. The increase in stress affects the process of material removal, which is mainly reflected in two aspects: first, it reduces the depth of the ductile brittle transition of the glass, and second, it is more likely to lead to brittle removal modes [65]. To understand the mechanism by which the previous scratch affects the crack behavior during the subsequent second scratch process, Feng et al. [66] designed an experimental tool, shown in Figure 6b, and conducted scratch experiments using the experimental equipment shown in Figure 6a under the experimental conditions of a load of 3 mN, scratch speed limit of 5 μm/s, and scratch length set to 100 μm. The distance between two scratches (*Ds*) and the crack behavior under secondary scratches were evaluated. Based on the different sizes of the *Ds*, the crack behavior can be divided into three stages, with only the early transverse crack stage occurring, the premature disappearance of transverse cracks without a stage of crack appearance, and the intermediate crack recurrence stage. As shown in Figure 6c–i, the crack behavior varied with the change in *Ds* (from 0.6 μm to 2.2 μm)

In subsequent studies, Feng et al. [67] sought to further investigate the propagation behavior of surface cracks on BK7 glass. To this end, they innovated the experimental tool by upgrading the traditional single-head tool to a double-head design. Scratch experiments were conducted with distances of 0.6, 0.8, 1.0, 1.2, 1.4, and 1.8 μm between the two tips (the scratch depths for each experiment were 200, 300, 400, 500, and 600 nm). The experiment showed that the double-tip scratches generated median intermediate cracks (MMCs) and edge transverse cracks (TLCs) due to stress superposition. As shown in Figure 7a, in the double-scratch experiment with a scratch depth of 500 nm, TLCs begin to disappear as the scratch depth increases from 1.4 μm to 1.6 μm. As shown in Figure 7b, in the double-scratch experiment with a scratch depth of 600 nm, MMCs are gradually replaced by MCs as the scratch depth increases from 1.2 μm to 1.6 μm. Figure 7c shows a summary of the crack changes in the experiment.

In the experiment, it was found that there was a significant difference in material damage between successive double scratching and double-tip scratching (the principle of successive double scratching and double-tip scratching is shown in Figure 8a,d). A comparison of Figure 8b,c,e,f reveals that the surface quality of the double-tip scratch is markedly superior to that of the successive double scratch, which facilitates the optimization of the material removal process. Gu et al. [23] studied the grinding process of BK7 glass. Through experiments, it was observed that under specific loading conditions, a single scratch can lead to the formation of cracks, but it does not significantly cause the phenomenon of blade collapse. However, in the double-scratch mode, the situation is different. In this mode, the surface morphology, material removal effect, and scratch depth are strongly influenced by two factors: the applied load and scratch spacing.

### 3.2. The Impact of the Scratch Parameters on the Surface of the BK7 Glass

The surface of the BK7 glass material is influenced by two primary factors: scratch depth and scratch speed [67]. In the processing of BK7 glass, the scratch depth (or cutting depth) plays a crucial role. Yu et al. [68] and Hu et al. [69] investigated whether the scratch depth (cutting depth) affects the removal mode of BK7 glass. As shown in Figure 9a–c, Yu et al. found that at a constant speed, adopting different cutting depths can lead to different removal modes of BK7 glass: toughness removal, toughness–brittleness removal, and brittleness removal. Figure 9d shows the transition of the BK7 glass removal mode with increasing depth at a constant scratch speed (Hu et al.).

In the processing of BK7 glass, although the scratch speed cannot directly determine the material removal method, it can change the proportion of different removal methods [68]. Feng et al. [70] conducted a study to examine the impact of scratch speed on the ductile–brittle transition (DBT) of BK7 glass. Scratch experiments were conducted on BK7 glass at 1 m/s, 5 m/s, and 20 m/s, as shown in Figure 10. In Figure 10a,a1,b,b1, when the scratch speed increases from 1 m/s to 5 m/s, DBT experiences a delay. This is mainly because the scratch process is highly synchronized with the thermal expansion phenomenon, resulting in effective relief of residual stress. However, in Figure 10c,c1, a velocity of 20 m/s results in the occurrence of DBT at an earlier point in time than a velocity of 5 m/s. This is because the thermal effect on DBT becomes negative, and high-intensity shear flow accelerates the earlier transformation of DBT. Overall, an increase in the scratch speed not only improves machining but also increases the surface temperature of the material [71,72]. This results in the accumulation of residual heat generated by continuous scratches (at high speeds), which in turn affects the DBT of the material.

Scratch experiments can help people better understand the changes that occur in the material itself during processing, but currently, most scratch experiments are simple and have some differences from the actual machining process. Most scholars, when studying the removal mechanism of BK7 glass, have focused on the influence of a single factor (such as scratch speed, number of scratches, or temperature). However, during machining, these factors coexist and interact with one another to affect the final result. In future research, more attention should be given to the removal mechanism of BK7 glass under the simultaneous action of multiple factors.

## 4. Research on Improvement Methods for the Surface Quality of BK7 Glass

### 4.1. Number of Studies on Different Machining Techniques for BK7 Glass

The significance of machining technology for the surface quality of BK7 glass lies in achieving efficient and precise machining, ensuring that the surface quality and shape accuracy meet the requirements. This article used Web of Science to search for articles on glass machining from May 2001 to the present, using machining technology and glass as the search criteria. A statistical chart was created of the classification of machining techniques, shown in Figure 11. According to the statistical chart, the use of grinding technology for glass machining has always been mainstream among researchers. This section primarily concerns the effect of grinding machining on the surface quality of BK7 glass.

The grinding wheel used for machining BK7 glass is usually composed of two parts: abrasive grains and an adhesive for fixing the abrasive grains [1]. During the grinding process, due to the high hardness and brittleness of BK7 glass [73], debris generated on the material surface gradually accumulates and adheres to the grinding surface of the grinding wheel. The accumulation of such debris not only affects grinding efficiency but also has a significant impact, which is that it gradually hardens the cutting edge of the abrasive particles. The originally smooth cutting edge becomes increasingly rougher after hardening, which increases the contact area and contact pressure with the workpiece surface, requiring greater force to drive the grinding wheel for grinding and generating higher grinding temperatures. This causes thermal damage to the material surface, resulting in surface damage and irreversible wear on the grinding wheel, reducing its service life [74,75,76,77,78]. To enhance the efficiency of grinding operations and guarantee the production of defect-free, high-quality glass surfaces, a comprehensive investigation of the grinding performance of grinding wheels is important.

### 4.2. Important Methods to Improve the Grinding Quality of BK7 Glass

There are two main approaches to improving the surface quality of BK7 glass: the first involves the development or adaptation of new types of tools, and the second involves the use of novel machining techniques such as ultrasonic and laser machining. This section introduces methods of improving the surface quality of BK7 glass by optimizing the tool design used in the process. The performance parameters of the cutting tool, including its sharpness, hardness, and wear resistance, directly influence the surface roughness of the processed BK7 glass [79].

In the use of grinding wheels, the main problems are twofold: first, the high wear rate of fine sand in the grinding wheel, and second, the low material removal rate. These two issues have an impact on the performance and processing efficiency of grinding wheels [80,81]. To address this issue, Wu et al. [82] adopted a method of laser fine-tuning of the geometric shape of abrasive diamond particles using the laser device shown in Figure 12a. This technology optimizes the cutting tools in the grinding wheel and significantly improves the grinding performance of the wheel. The shape of the abrasive diamond particles after laser fine-tuning is shown in Figure 12b, exhibiting more optimized and refined geometric characteristics. When using this adjusted tool for grinding BK7 glass, the surface roughness of the obtained glass is relatively reduced. By observing and comparing Figure 12b,c, it is evident that the repaired grinding wheel has relatively less wear after grinding BK7 glass. The main wear is concentrated at the tip of the abrasive grains, which shows that quenched and tempered abrasive diamond grains have significant advantages in the precision grinding of BK7 glass. The internal microstructure of traditional hot-pressed sintered grinding wheels is relatively closed, with low porosity, which may not only lead to wheel blockage but also easily cause surface burns on the workpiece. As the grinding of traditional hot-pressed sintered wheels is mainly dependent on machining experience and lacks a targeted design, the performance is often unsatisfactory [76]. To address these issues, structured grinding wheels have been proposed as a new type of grinding performance control scheme. The grinding performance of grinding wheels is optimized by designing specific macroscopic and microscopic structures, including cooling macroscopic structures [75], surface microscopic structures [78], and pore structures, to achieve more accurate and efficient grinding machining. Wang et al. [18] used a digital design and combined 3D printing technology to manufacture five new types of grinding wheels with different microstructures, which had many interconnected pores and liquid channels. The shape, dimension, number, and distribution of the micropores can be designed and adjusted to fundamentally improve the problems of wheel blockage and surface scorching. Compared with electroplated grinding wheels, the new type of grinding wheel exhibits a lower cutting force and grinding energy when machining BK7 glass while improving the surface quality of the workpiece. Furthermore, this new type of grinding wheel is self-repairing, requires no secondary dressing, and has superior wear resistance, self-sharpening, and grinding performance.

The volume of BK7 glass material processed is usually not very large, and traditional grinding wheels have difficulty effectively machining it. Microgrinding technology [83,84] is a very effective means to achieve high-precision machining of hard and brittle materials. By finely controlling the grinding force and depth, it is possible to achieve the desired level of precision. Microgrinding is a technique that can be employed to achieve high-precision machining while simultaneously improving the surface quality of the parts in question. By removing defects such as surface burrs and rounded edges, the smoothness and flatness of parts can be improved, further improving their performance and service life. Pratap et al. [4] designed four types of microtextured tool structures, T2–T5, as shown in Figure 13a, for microgroove grinding of BK7 glass based on the working principle of EDM in Figure 13b. After experimental verification, it was observed that the cutting force generated using microtextured cutting tools was significantly lower than that generated using ordinary commercial tools. The primary rationale for this phenomenon is that the microtexture structure of the tool facilitates the discharge of debris and the flow of lubricant, thereby reducing friction and reducing tool wear. This indicates that microtextured tools are superior to ordinary tools in BK7 glass microgrinding, providing a new method for microgrinding.

In subsequent research, researchers adjusted the discharge machining parameters to modify diamond cutting tools and found that the height of the abrasive protrusions and cutting depth increased under high line tension, leading to an increase in cutting force [85]. In the process of studying microtextured tools, it was found that the size of the tool texture and the number of texture units also have an impact on grinding performance. Tools with large texture units have a greater penetration depth, while tools with smaller texture units may delay the material’s transition from plastic to brittle [86]. Pratap et al. [87] considered the interaction between the tool and the workpiece surface and established a cutting force analysis model for BK7 glass microgroove grinding, with fewer errors than those of the Kadivar [88] model. Afterward, as shown in Figure 13c, they produced better-designed microtextured tools [89]. Through this type of tool, tools with micropool arrays were produced, as shown in Figure 13d,e. Experiments have shown that compared with previous microgroove cutting tools [4], tools designed with micropool array textures exhibit lower cutting forces during the cutting process, which directly proves their excellent grinding performance. This not only highlights the significant advantages of microtexture design in reducing cutting resistance and improving machining efficiency but also further emphasizes the important value of microtextures in improving material surface quality.

**Figure 13 micromachines-15-01021-f013:**
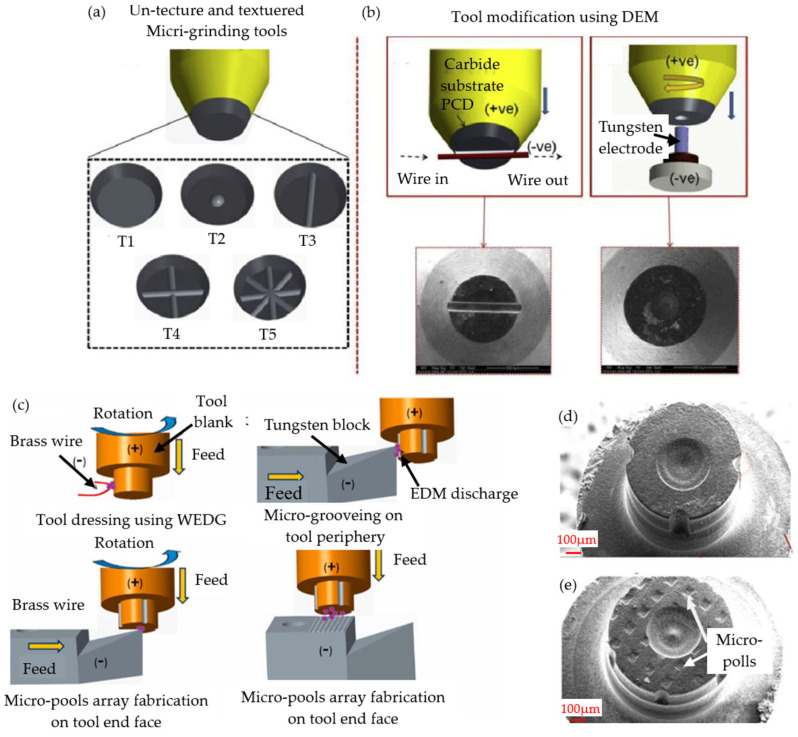
Design of microtextured cutting tools: (**a**) tool design [4]; (**b**) EDM cutting tools [4]; (**c**) tool correction machining [89]; (**d**,**e**) finished cutting tools [86,89].

## 5. Subsurface Damage (SSD) of BK7 Glass

### 5.1. Influence of SSD on the Optical Properties of BK7 Glass

When BK7 glass material is ground using inappropriate machining parameters, brittle damage often occurs on the workpiece [90], resulting in microcracks on the subsurface, which are called subsurface damage [91]. The SSD usually decreases with increasing depth, as shown in the subsurface damage distribution in Figure 14a. Figure 14b,c show the common subsurface damage, which follows the distribution pattern mentioned above. SSD can seriously affect the surface integrity and mechanical and optical properties of BK7 glass, thereby decreasing the service life and long-term stability of the material. In the use of optical components, under the conditions of field enhancement caused by interference, enhanced absorption caused by material captured in cracks, and increased mechanical weaknesses, subsurface cracks can affect the sensitivity of laser damage and cause macroscopic damage to optical components [92]. Although SSD has an impact on the performance of BK7 glass, its detection and evaluation become quite difficult due to its occurrence below the material surface. At present, many methods for evaluating SSD inevitably change or even damage the surface of BK7 glass and require the use of costly equipment and highly skilled professionals or extensive preparation work [93]. Many scholars have actively investigated an efficient and accurate evaluation method for the subsurface stress distribution (SSD) problem in BK7 glass. This is not only crucial for a deeper understanding of the generation mechanism of SSD but also has profound implications for the design optimization, manufacturing process improvement, and performance enhancement of BK7 glass in practical applications.

### 5.2. A Predictive Model for SSD of BK7 Glass

The evaluation methods for SSD can be roughly divided into two categories: destructive and nondestructive. Destructive methods involve the application of additional treatments to the materials under examination, with the objective of exposing the SSD and rendering it visible and amenable to measurement. Nondestructive methods evaluate SSD without damaging optical materials, but they have high requirements of operators, rely on expensive equipment, and are sometimes unable to provide accurate and reliable SSD data. To more effectively evaluate SSD, scholars are investigating the clear correlation between SSD and surface roughness (SR). By clarifying this correlation and utilizing the actual measured SR values, a mathematical model that can predict SSD was established. Li et al. [95] modeled the surface and SSD caused by grinding machining using cup-shaped grinding tools. Through experimental sample validation and analysis, it was found that the evaluated deviation of the model was very small when BK7 glass was processed in both ductile and DBT modes. However, in the brittle mode, the high randomness of brittle material removal modes led to an increase in evaluation deviation.

However, these evaluation methods have shown some limitations in the on-site monitoring of subsurface losses. Therefore, some researchers have conducted in-depth explorations of the relationship between the cutting force and machining surface depth during the diamond cutting process and proposed a nonlinear theoretical correlation between them. Furthermore, real-time monitoring of the cutting force can predict the depth of subsurface cracks, achieving online monitoring and optimization of machining quality [94]. Lv et al. [96] combined statistical probability analysis of abrasive height with the indentation fracture principle of brittle materials to study the correlation between the cutting force of diamond tools and the maximum subsurface crack depth in ultrasonic machining. They successfully constructed a theoretical prediction model to explain the relationship between SSD and cutting force measurements. This model is capable of accurately predicting the SSD depth of machined surfaces based on key parameters such as the cutting force of diamond cutting tools, mechanical properties of the processed material, and geometric properties of abrasives. At the same time, it also has the capacity to monitor the depth of the SSD in real time and online throughout the machining process. Nevertheless, the aforementioned methodologies fail to consider the impact of elevated strain rates and elevated temperatures experienced by the workpiece surface during grinding machining. Brittle materials have strain rate enhancement effects [97,98,99], and high strain rates (>10^3^ s^−1^) are the main cause of material embrittlement. Qi et al. [28] investigated how the strain rate and temperature affect the grinding quality of BK7 glass. The objective of that study was to gain insight into the influence of changes in the speed of individual abrasive scratches on the grinding process of BK7 glass. On this basis, a subsurface damage depth prediction model for single abrasive scratches on BK7 glass was successfully constructed. Under constant cutting depth conditions, as the strain rate and temperature in the contact area of the abrasive workpiece gradually increase, a decreasing trend is observed in the radius of the plastic deformation zone, which hinders the generation and propagation of intermediate cracks, transverse cracks, and radial cracks. This phenomenon is advantageous for reducing crack length, preventing brittle fracture, enhancing surface integrity, and mitigating underground damage.

During the machining process, a noteworthy phenomenon is that the local temperature levels experienced by the chip formation area and abrasive particles may significantly exceed the actual surface temperature of the workpiece or even reach several times the latter. The mechanical properties of optical glass materials significantly change with increasing pressure head loading and temperature. Li et al. [100] successfully developed a comprehensive model based on the consideration of the local strain rate and temperature in the grinding area combined with the theoretical framework of grinding kinematics. The impact of the strain rate and grinding temperature on the mechanical properties and grinding performance of optical glass was systematically analyzed. With the increase in the local strain rate in the grinding area and the increase in the circumferential temperature during the grinding process, the fracture toughness of the material increases. However, at the same time, the microhardness decreases, and the SSD also decreases. The influence of the local strain rate and temperature on the damage mechanism of BK7 glass under different grinding wheel speeds (*Vs* = 30, 60, 90, 120 m/s) was investigated, as shown in Figure 15a. The depth of subsurface loss of BK7 glass after machining was found to be significantly reduced with an increase in grinding wheel speed. This improvement in the surface quality of the material was attributed to the reduction in subsurface damage. Figure 15b shows that the subsurface morphology of the ground BK7 glass consists mainly of shell-shaped defects due to radial cracks, with transverse cracks, and intermediate cracks appearing. During the entire grinding process, as the grinding wheel speed gradually increases, the size of the defects appearing on the shell significantly decreases.

## 6. Ultrasound-Assisted Machining (UAM)

The grinding process is one in which mechanical vibration plays a pivotal role. Its influence on the surface quality of the final product cannot be overlooked. The vibration of the spindle system and the setting of grinding parameters are considered the main sources of precision grinding errors [101,102]. Figure 16 shows that regardless of the use of tools and machining conditions, the superimposed ultrasonic vibration slightly decreases the subsurface quality [103]. During the machining of BK7 glass, the effect of ultrasonic vibration on the surface and subsurface quality is a critical factor that cannot be ignored. Research has shown that the use of ultrasound-assisted machining can effectively improve the surface and subsurface quality of workpieces [104,105,106].

The essence of UAM technology lies in its ability to utilize the high-frequency vibration of ultrasound to act on the surface of workpieces. This high-frequency vibration not only changes the material removal mechanism on the surface of the workpiece through physical means but also affects the properties of the workpiece material to a certain extent, thereby accelerating the machining process and improving the machining quality. Ultrasonic machining has demonstrated its unique and important value in the field of brittle material machining [107]. This machining method is commonly used in various material machining fields, including for metals, plastics, ceramics, and composite materials. UAM has many advantages, such as high machining accuracy [108], fast speed, wide adaptability of machining materials, and the ability to process complex cavities and surfaces. Ultrasonic machining technology has shown significant advantages in machining brittle materials. Due to the high-frequency vibration generated by ultrasonic waves, the contact between the tool and the workpiece becomes very slight, resulting in a significant reduction in cutting force [109]. The characteristics of light contact and small cutting force make the machining process smoother and more efficient and effectively avoid workpiece damage caused by excessive cutting force. In addition, the structure of ultrasonic machining machines is simple and easy to maintain. Due to their excellent processing performance, ultrasonic machines can be used for precision grinding of BK7 glass, which not only improves the processing efficiency and reduces tool wear [110] but also reduces surface roughness and reduces the removal margin and time of subsequent polishing machining. In addition, ultrasonic machining technology plays a crucial role in improving precision machining theory and promoting the development of high-tech cutting-edge technologies [111,112,113,114]. This article mainly introduces two types of machining in UAM, namely, ultrasonic vibration-assisted grinding (UAG) and ultrasonic vibration-assisted polishing (UVAP).

### 6.1. Ultrasonic Vibration Assisted Grinding (UAG)

UAG is a composite material machining technology that combines traditional grinding processes with advanced ultrasonic vibration processing technology. It utilizes the oscillation effect of ultrasound to improve physical phenomena (such as friction, deformation, and cutting force) in traditional grinding machining, thereby improving grinding efficiency and quality [115]. The two common machining devices for UAG are shown in Figure 17a,b, where it can be observed that the main difference between them is the presence or absence of a cooling system. The main difference compared to traditional grinding devices is the addition of an ultrasonic generator. This results in a number of significant advantages, including a reduction in grinding force and temperature, an extension of tool life, an improvement in the material removal rate, an enhancement in the surface and subsurface quality of the workpieces, and a suppression of wheel blockage [116,117]. Therefore, the construction of a surface roughness prediction model and a cutting force model represents a pivotal step in the process of gaining a deeper understanding of the mechanism by which UAG reduces grinding forces and improves grinding quality.

In the machining of UAG, there are three main forms of mechanical motion: rotational motion of the spindle, which is the basic motion in grinding machining; feed motion along the cutting direction, which is responsible for pushing the workpiece or grinding tool for material removal; and the vibration motion of the ultrasound along the axis, which utilizes the high-frequency vibration of the ultrasound to improve grinding efficiency and quality. Huang et al. [2] considered the nonuniformity of the protrusion height of grinding wheel abrasives and established a theoretical model for the cutting force of BK7 glass with an UAG. Through model analysis, it was found that adjusting certain grinding parameters can significantly affect the magnitude of the cutting force. Specifically, an increase in spindle speed, a reduction in cutting depth, and a decrease in feed rate all contribute to a reduction in cutting forces. In addition, the model also shows that as the amplitude of ultrasonic vibration increases, the cutting force of the UAG shows a nonlinear downward trend. To achieve an accurate prediction of surface roughness for brittle materials machined with an UAG, Zhao et al. [118] studied the specific effects of grinding wheel morphology and processing parameters on the trajectory of abrasive particles in the UAM process of glass materials. This study established a theoretical surface roughness model. Brittle materials are prone to undergo fracturing during processing, which has a significant effect on the surface quality of the process, and therefore a Gaussian machining regression method was employed to establish the model.

In subsequent studies, they [119] investigated the influence of processing parameters on the material removal mechanism. As shown in Figure 18, grinding experiments were conducted using feed rate (*V_f_* = 10,110 mm/min), depth of cut (*a_p_* = 10, 80 μm), and amplitude (A = 7 μm). Figure 18a–d show the relationship between feed rate and the number of brittle fracture pits. Specifically, as the feed rate gradually increases, the number of brittle fracture pits also increases. A comparison of Figure 18b,e clearly reveals that as the grinding depth increases, the number of large brittle fracture pits also increases. Following the introduction of ultrasonic vibration into the grinding process, significant changes were observed in the material removal mechanism, as shown in Figure 18c. Figure 18d shows that when the ultrasonic amplitude is 7 μm, the number of large brittle fracture pits decreases because they transform into ductile removal pits. During ultrasonic machining, the material is subject to the comprehensive influence of abrasive particles and the random distribution of size and penetration depth. Lv et al. [120] paid special attention to the kinematic characteristics and tool surface distribution characteristics during the processing of BK7 glass rotary ultrasonic machining technology and studied the dynamic changes in cutting force. Subsequently, they integrated the Gaussian distribution characteristics of grain size and penetration depth to construct a novel theoretical model for the prediction of cutting forces.

### 6.2. Ultrasonic Vibration-Assisted Polishing (UVAP)

UVAP utilizes high-frequency vibrations generated by ultrasonic generators [121,122], which are converted into mechanical vibrations through transducers and transmitted to polishing tools or workpieces. Figure 19a shows a schematic diagram of UVAP, and Figure 19b shows the actual UVAP device. High-frequency vibration has two principal effects during polishing and machining. First, ultrasonic machining technology effectively improves the material removal rate by increasing the relative speed of motion between the polishing tool and the workpiece. An increase in the speed allows abrasive particles to interact more frequently with the surface of the workpiece, thereby improving the processing efficiency. Second, the technology also utilizes the cavitation and impact effects of ultrasound. These effects can significantly enhance the contact and friction between abrasive particles in the polishing solution and the surface of the workpiece, making the removal of surface materials more uniform and efficient. This not only helps to improve the surface smoothness of the workpiece but also further improves its overall quality. Compared with traditional polishing, UVAP has a higher machining efficiency, better surface quality, and wider applicability.

Ultrasonic polishing can be understood in abstract terms as the high-frequency impact of a multitude of abrasive particles against the surface of brittle materials, which removes the surface material at a microscopic level, leaving the surface flat and free of defects [125,126]. The polishing machining process involves the removal of material, which is a complex phenomenon. The mechanism primarily relies on the wear caused by abrasive particles, which can be either fixed or loose. When studying the ultrasonic polishing process, to more accurately reveal the correlation between the abrasive particle distribution and material removal efficiency, researchers introduced the concept of the “Preston coefficient distribution function” [127]. This function has been designed as a quantitative tool to describe the polishing particle dispersion state during the polishing process and how this dispersion state affects the material removal function [128]. In a study published by Zhang et al. [124], a distribution function model for the removal of UVAP material was proposed. This model was based in part on the Preston equation, which takes into account the periodic changes in polishing force and contact radius that occur as a result of ultrasonic vibration. This model deeply analyzes the role of ultrasonic vibration in the polishing process and clearly indicates that the polishing ability is greatly improved by introducing ultrasonic vibration. Furthermore, the model reveals that an increase in axial ultrasound amplitude can further enhance this polishing ability. For purposes of understanding the removal performance for brittle materials during UVAP, Zhang et al. [123] developed a BK7 glass material removal model using the Preston equation, which has greater accuracy in predicting material removal rates than Zhang’s model [124]. There is a direct correlation between the maximum material removal depth and the spindle speed in combination with the ultrasonic amplitude. By applying this model, multiple key parameters during the machining process can be reasonably predicted. To enhance the material removal rate and machining efficiency, Liang et al. [129] proposed oblique angle polishing of BK7 glass and developed a model for the material removal process. The polishing angle has a significant impact on the accuracy of the grating angle profile and material removal rate [130]. This provides new ideas for understanding the mechanism of UVAP and improving machining efficiency. To investigate the impact of machining parameters on vibration polishing, Mounir H et al. [131] examined the influence of vibration frequency, polishing time, and polishing machine type on the final state of BK7 glass surfaces. An experimental comparison was conducted under the polishing conditions shown in Table 1. The material removal rate displayed in Figure 20a is positively correlated with the working time. This phenomenon can be attributed to the continuous presence of abrasive particles, which provide power for polishing machining. The different material removal rates between the two polishing machines can be explained by the different void rates of the polishing machines. A high porosity is conducive to the transmission and storage of polishing fluid, thereby improving the polishing effect and material removal rate. Figure 20b clearly shows that the surface roughness of the BK7 glass is negatively correlated with time, and the surface roughness tends to remain unchanged at 15 min. The aforementioned research serves as a reference point for polishing machining.

## 7. Conclusions and Future Research Trends

With the continuous progress of optical technology and the ceaseless expansion of application fields, the demand for BK7 glass has shown an increasingly diverse trend. Whether in high-precision optical instruments, medical devices, or advanced optoelectronic equipment and communication systems, BK7 glass is highly favored for its excellent physical and chemical properties. In light of the accelerated advancement in optical technology and the continuous expansion of application fields, BK7 glass, a key optical material, must be able to meet the specific needs of different fields and diverse application scenarios. This diversified demand not only greatly promotes the prosperity and development of the BK7 glass industry but also brings unprecedented challenges to its processing and manufacturing. At present, research on BK7 glass still faces many difficulties, including the hard and brittle properties of the material, strict optical performance requirements, machining and polishing technology challenges, raw material and cost issues, and environmental and sustainability requirements. Therefore, to promote the development of BK7 glass machining and manufacturing and meet its growing application needs, it is particularly important to establish a comprehensive theoretical analysis model, improve existing machining and manufacturing technologies, and explore efficient machining routes. Based on the current research status of BK7 glass and its application needs in various fields, future research should focus on the following key aspects.

The machining technology used for BK7 glass has a significant impact on its performance and application. The main research direction for improving machining technology in the future is continuous innovation and optimization. On the one hand, multiple machining techniques can be combined to improve the machining accuracy and surface quality of BK7 glass, reduce damage and defects during processing, significantly improve the machining accuracy and surface quality of BK7 glass, and ensure high product quality. On the other hand, actively exploring and developing new machining technologies and equipment, such as using artificial intelligence and machine learning to optimize machining parameters and process control and achieving intelligent and automated machining, can not only reduce production costs but also improve the machining efficiency of BK7 glass, meeting the growing demand for high-quality and efficient glass products in the market.Microtextured cutting tools play an important role in machining and can improve cutting performance, optimize chip control, enhance tool adaptability, and improve machining accuracy and surface quality. In subsequent research, two aspects can be considered: microtexture design and microtexture machining. For microtexture design, the application of different shapes and directions of microtextures in BK7 glass machining can be explored to achieve specific machining effects and performance requirements. In terms of microtexture machining, composite processing can be considered, combining various machining techniques such as electric discharge machining and laser machining to achieve composite machining of microtexture structures and achieve better machining results.Ultrasound-assisted machining, a cutting-edge technology, has broad application prospects. In the future, the interaction mechanism between UAM and grinding parameters can be further studied. By analyzing how these parameters affect each other and their specific effects on the machining process, the optimal combination of machining parameters can be explored. Completing machining tasks in a shorter time while ensuring better surface quality to meet the high-precision and quality requirements of BK7 glass in optical, medical, and other high-end applications is the primary goal. Moreover, by introducing advanced ultrasonic vibration generators and control systems, precise control of ultrasonic vibration parameters can be achieved, thereby further improving the stability and accuracy of machining.In future research on BK7 glass machining and manufacturing, the combination of microtextured cutting tools and UAM is undoubtedly an important and promising direction. The design and optimization of microtextured cutting tools can significantly improve chip removal efficiency and lubricant flow performance, ensuring a smoother cutting process. Ultrasonic vibration technology can improve the grinding ability and material removal rate of cutting tools. These two technologies should be combined to achieve efficient machining of BK7 glass.To promote the continuous progress of BK7 glass grinding technology, interdisciplinary and interdisciplinary cooperation and communication must be strengthened. By integrating knowledge and technological resources from multiple fields, such as materials science, mechanical engineering, control engineering, artificial intelligence, and environmental science, a comprehensive innovation system can be formed. For example, research achievements in materials science and developing new abrasives and grinding tools suitable for BK7 glass can be combined to overcome the challenges in this field.

## Figures and Tables

**Figure 1 micromachines-15-01021-f001:**
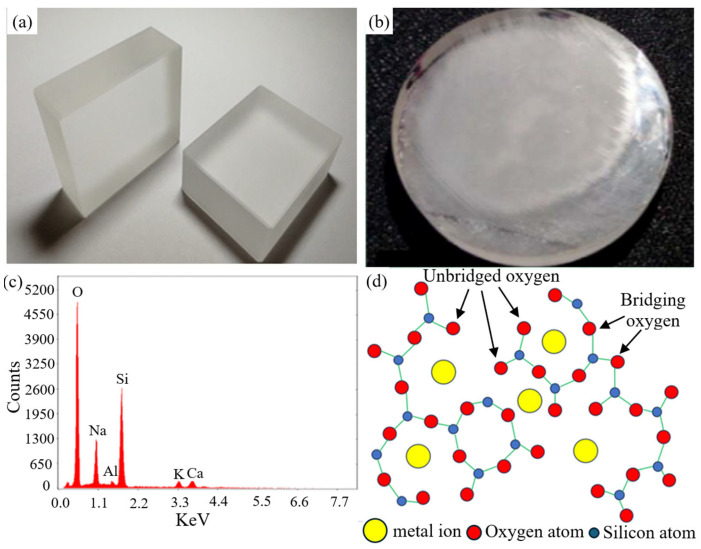
Morphology and elemental composition of BK7 glass: (**a**) macrostructure of BK7 glass [18]; (**b**) surface morphology after polishing [19]; (**c**) energy spectrum of BK7 glass [18]; (**d**) atomic structure diagram of BK7 glass.

**Figure 2 micromachines-15-01021-f002:**
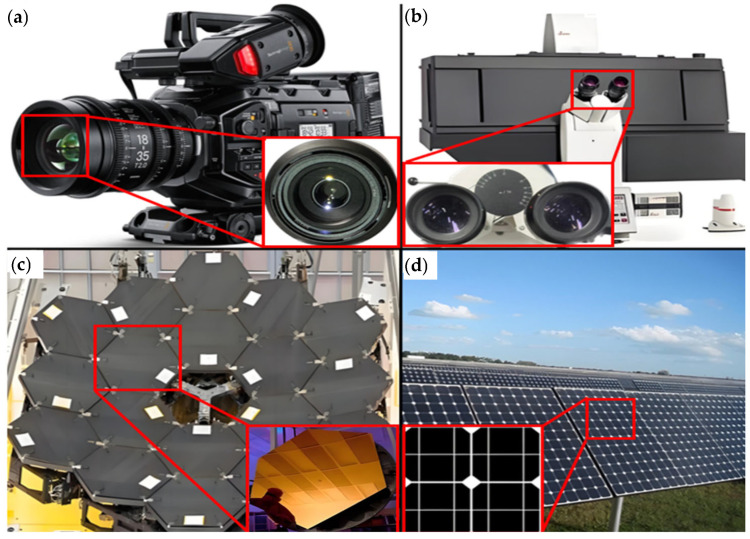
Application of BK7 glass: (**a**) application of BK7 glass in cameras; (**b**) application of BK7 glass in optical microscopes; (**c**) application of BK7 glass in space telescopes; (**d**) application of BK7 glass in solar panels.

**Figure 4 micromachines-15-01021-f004:**
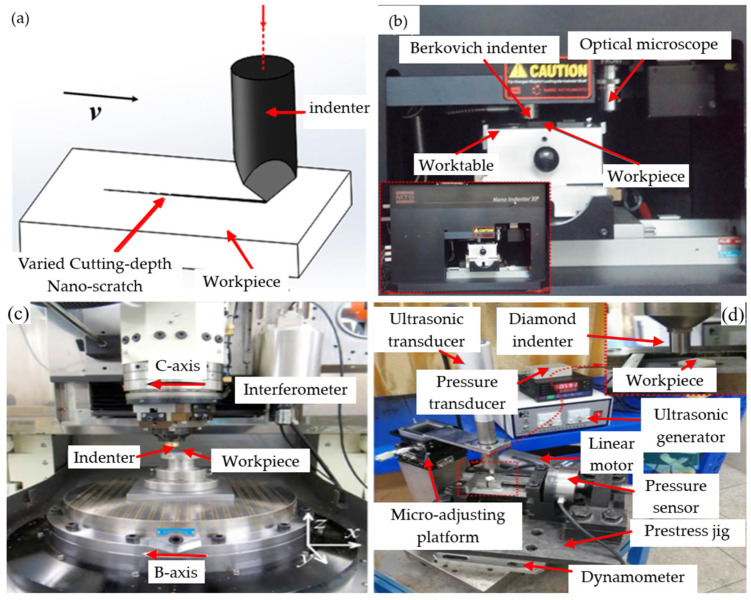
Principle and device diagram of scratch experiment: (**a**) typical experimental principles of scratch experiment [48]; (**b**) nanoscratch device [48]; (**c**) single abrasive scratch device [54]; (**d**) ultrasonic scratch device [55].

**Figure 5 micromachines-15-01021-f005:**
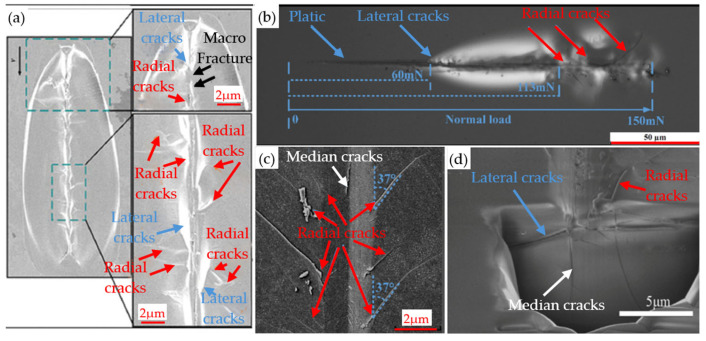
The three common types of cracks are lateral: (**a**) lateral cracks and radial cracks [57]; (**b**) lateral cracks and radial cracks [58]; (**c**) median cracks and radial cracks [58]; (**d**) lateral cracks, median cracks, and radial cracks [59].

**Figure 6 micromachines-15-01021-f006:**
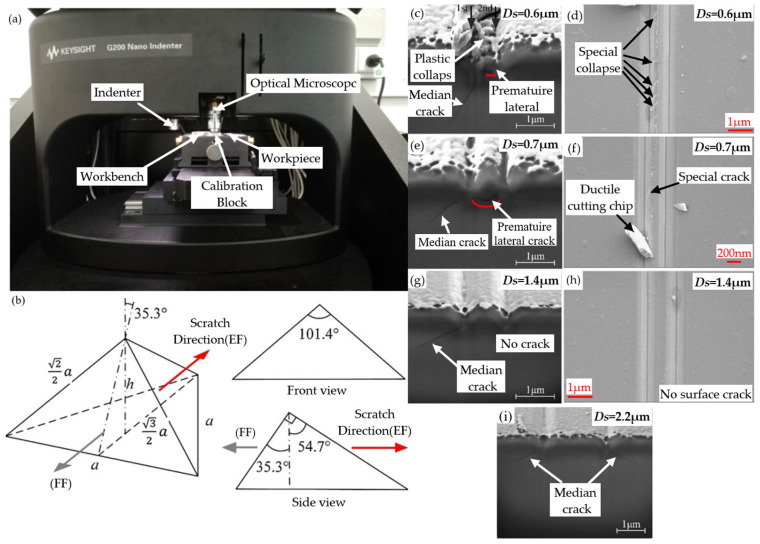
Double-scratch experiment: (**a**) experimental equipment [57]; (**b**) tool diagram [66]; (**c**–**i**) crack behavior changes in *Ds* from 0.6 μm to 2.2 μm [66].

**Figure 7 micromachines-15-01021-f007:**
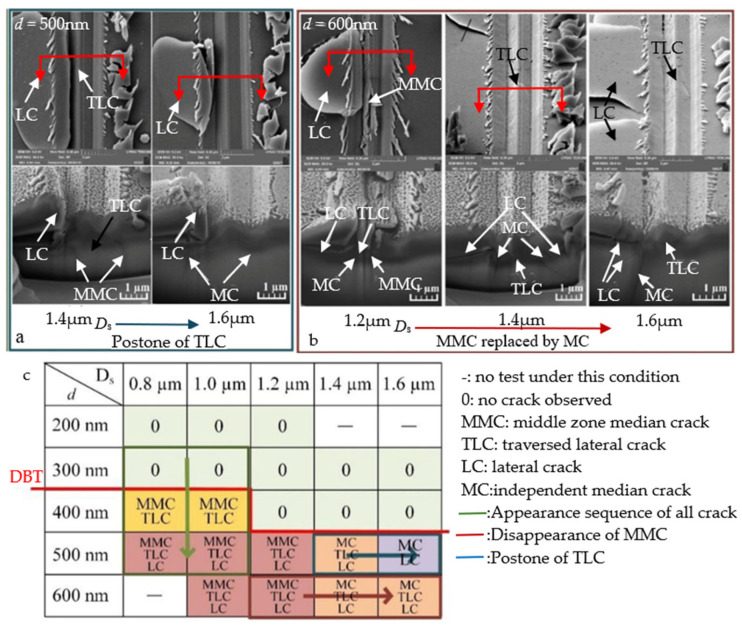
Changes in MMC and TLC [67]. (**a**) A series of scanning electron microscopy (SEM) images were obtained in a double-scratch experiment with a scratch depth of 500 μm. These images were captured as the scratch depth increased from 1.4 μm to 1.6 μm. (**b**) A series of SEM images were obtained in a double-scratch experiment with a scratch depth of 600 μm. These images were captured as the scratch depth increased from 1.2 μm to 1.6 μm. (**c**) Summary of crack changes in the experiments.

**Figure 8 micromachines-15-01021-f008:**
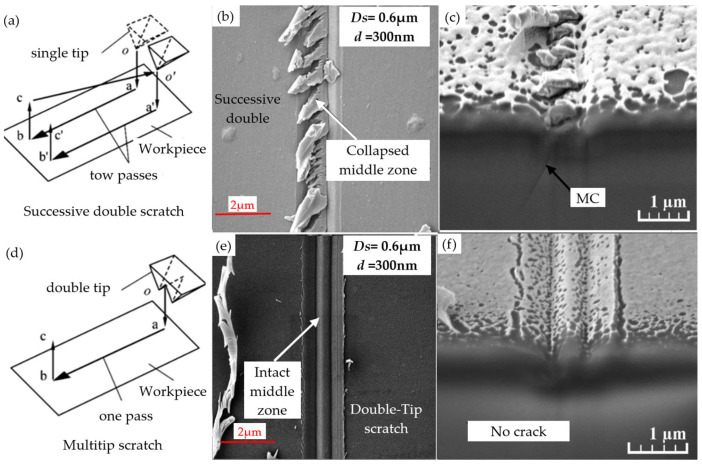
Successive double scratches and double-tip scratches [67]. (**a**) Schematic diagram of successful double scratches. (**b**) Surface quality of successive double scratches. (**c**) Continuous double-scraping cross-sectional morphology. (**d**) Schematic diagram of double-tip scratches. (**e**) Surface quality of double-tip scratches. (**f**) Cross-sectional morphology of double-ended scratches.

**Figure 9 micromachines-15-01021-f009:**
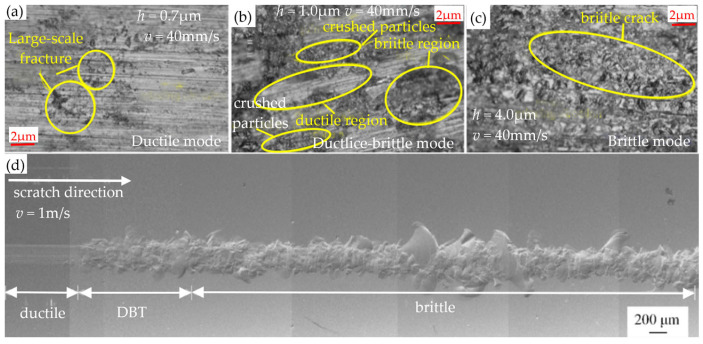
The impact of scratch depth (cutting depth) on the removal mode of BK7 glass. (**a**) *h* = 0.7: μm: ductile mode [68]. (**b**) *h* =1.0 μm: ductile–brittle mode [68]. (**c**) *h* = 4.0 μm: brittle mode [68]. (**d**) The transition of BK7 glass removal mode (*v* = 1 m/s) as the scratch depth increases [69].

**Figure 10 micromachines-15-01021-f010:**
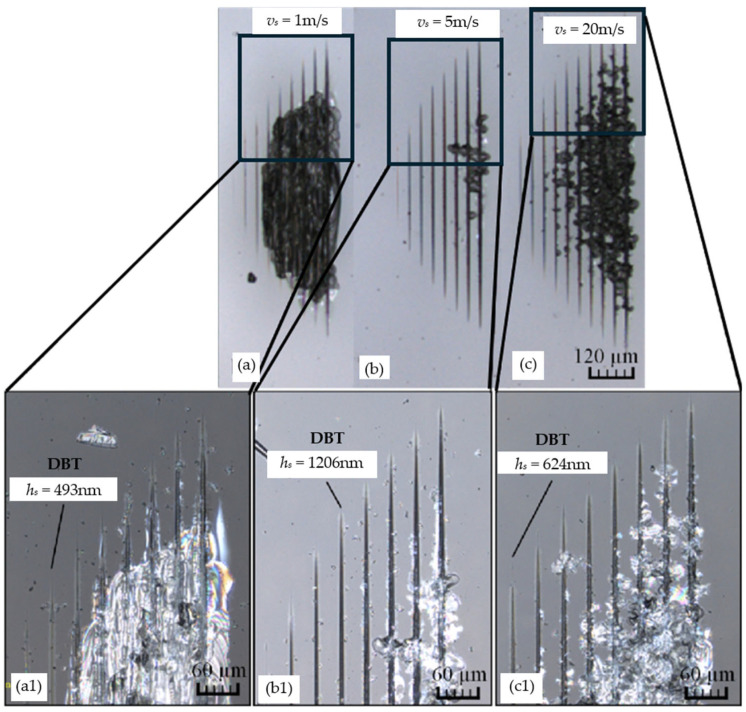
Influence of scratch speed on BDT [70]: (**a**,**a1**) BDT state at a scratch speed of 1 m/s; (**b**,**b1**) BDT state at a scratch speed of 5m/s; (**c**,**c1**) BDT state at a scratch speed of 20 m/s.

**Figure 11 micromachines-15-01021-f011:**
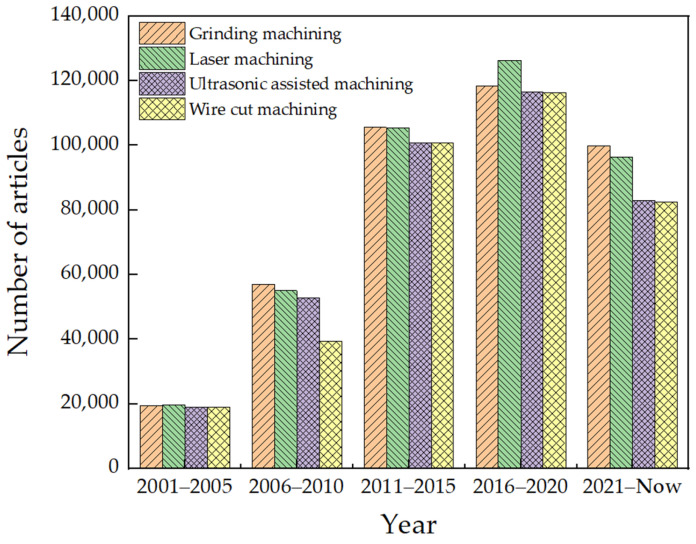
Number of publications (using Web of Science to search for literature).

**Figure 12 micromachines-15-01021-f012:**
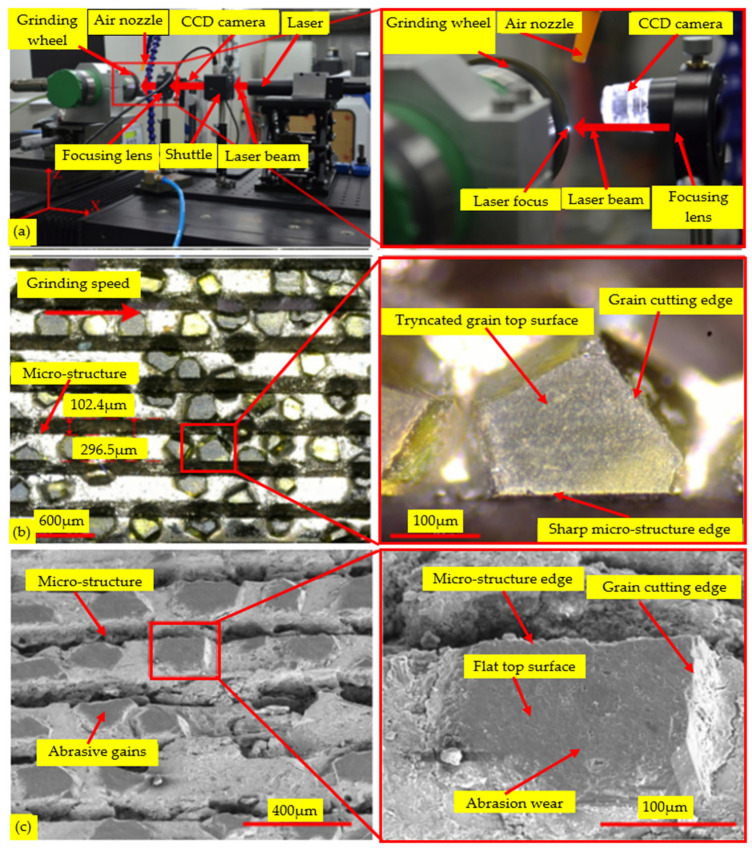
Optimization of grinding wheels [82]. (**a**) Laser dressing device. (**b**) Laser used to adjust the shape of abrasive particles. (**c**) Shape of the wear on abrasive particles resulting from the use of grinding wheels.

**Figure 14 micromachines-15-01021-f014:**
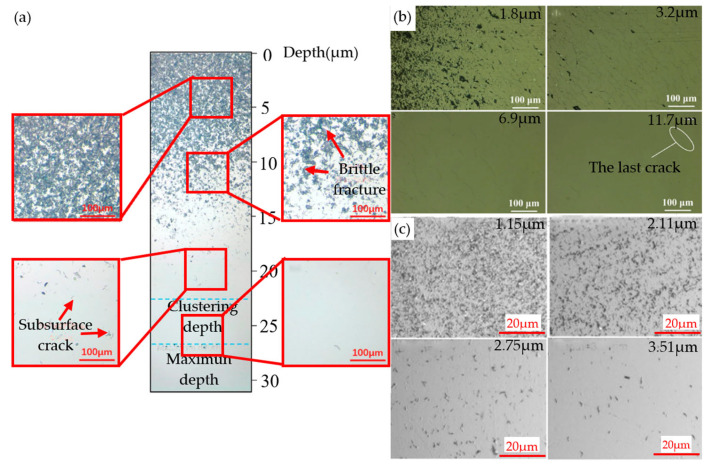
Subsurface damage distribution and common subsurface damage: (**a**) distribution of subsurface damage resulting from grinding at varying depths [90]; (**b**,**c**) common subsurface damage [91,94].

**Figure 15 micromachines-15-01021-f015:**
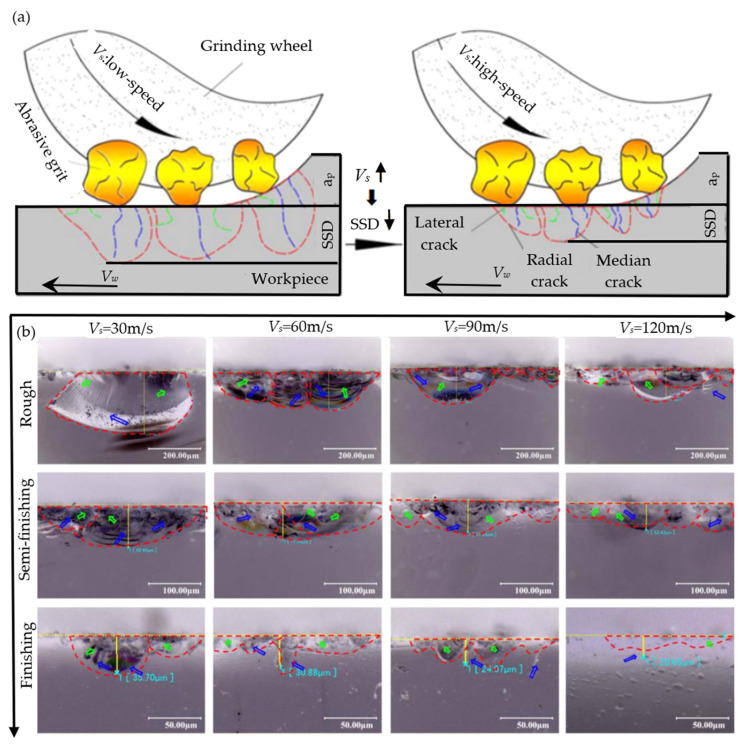
The effect of the strain rate on SSD [100]. (**a**) Schematic diagram illustrating the depth evolution of an SSD with respect to the impact of the wheel speed. (**b**) Relationship between the morphology of the SSD and the grinding wheel speed at different grinding stages. The red dashed line represents shell-shaped defects, while the green arrow and blue arrow represent transverse cracks and intermediate cracks, respectively.

**Figure 16 micromachines-15-01021-f016:**
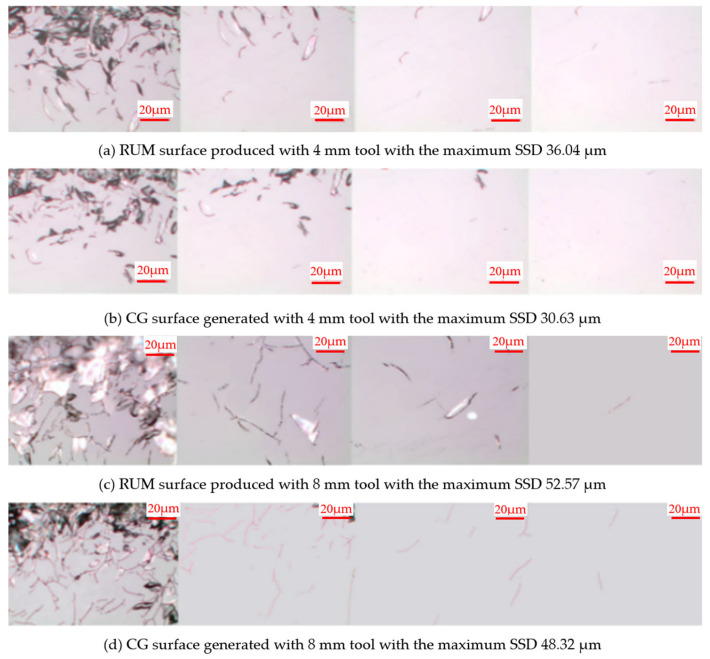
Typical SSD micrographs of specimen surfaces generated with and without ultrasonication [103]; (**a**,**b**) Subsurface damage caused by grinding with and without ultrasonic assistance using a 4 mm tool; (**c**,**d**) Subsurface damage caused by grinding with and without ultrasonic assistance using a 8 mm tool.

**Figure 17 micromachines-15-01021-f017:**
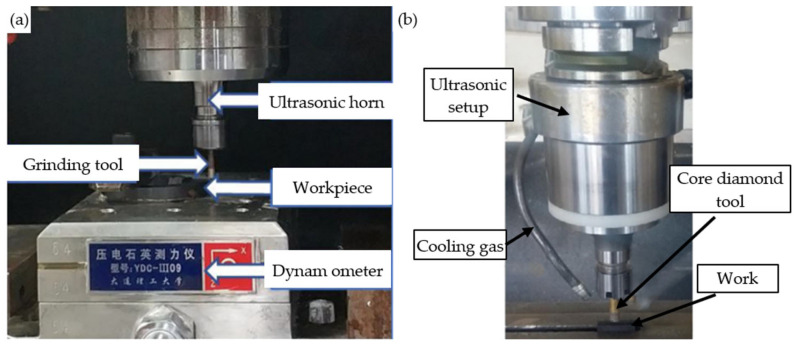
Ultrasonic Vibration Assisted Grinding device: (**a**) Ultrasonic Vibration Assisted Grinding device without cooling system [112]; (**b**) Ultrasonic Vibration Assisted Grinding device with cooling system [113].

**Figure 18 micromachines-15-01021-f018:**
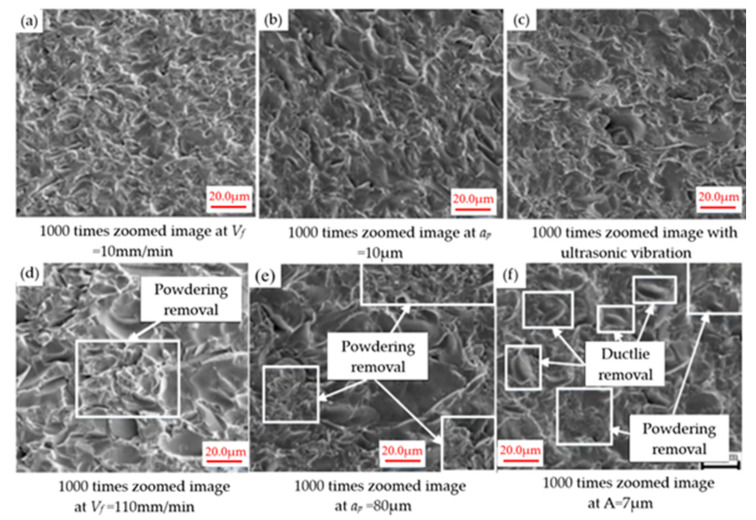
The impact of machining parameters on the removal of materials and the surface morphology characteristics [119]. (**a**) Operating at a feed rate of 10 mm/min. (**b**) At a cutting depth of 10 μm. (**c**) Adding ultrasonic vibration. (**d**) Operating at a feed rate of 110 mm/min. (**e**) At a cutting depth of 80 μm. (**f**) At an amplitude of 7 μm.

**Figure 19 micromachines-15-01021-f019:**
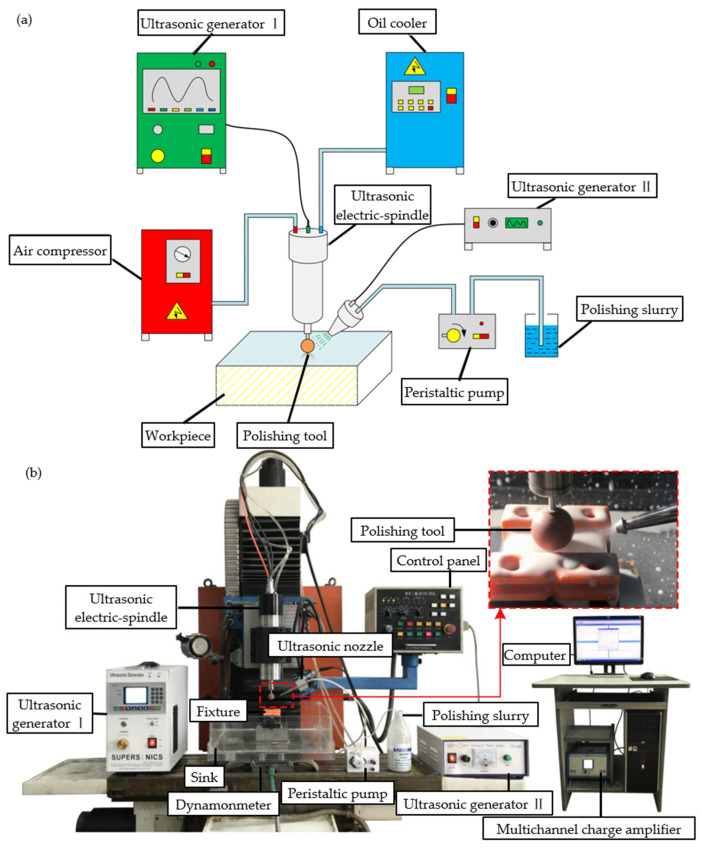
Ultrasound-assisted polishing: (**a**) schematic diagram of the UVAP machine [123]; (**b**) physical image of the UVAP machine [124].

**Figure 20 micromachines-15-01021-f020:**
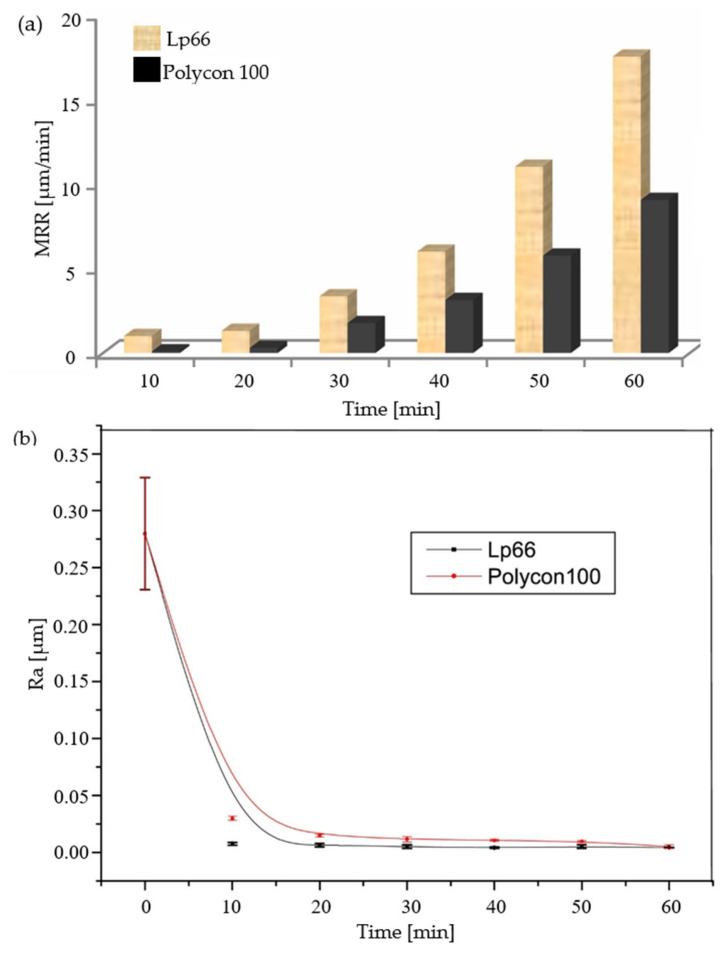
The rate of material removal from polished BK7 glass and the fitting curve of the surface roughness of BK7 glass [131]: (**a**) the rate of material removal from polished BK7 glass; (**b**) the fitting curve of the surface roughness of BK7 glass.

**Table 1 micromachines-15-01021-t001:** Polishing conditions [131].

Polishing Conditions			
Time (min)	10, 20, 30, 40, 50, 60	Frequency	4.62 kHz
Polisher	Lp 66, Polycon 100	Sample	BK7 glass Ø20 mm

## Data Availability

Not applicable.

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
