# Peer review of "Comprehensive Review on Research Status and Progress in Precision Grinding and Machining of BK7 Glasses"

_micromachines, 2024, doi:10.3390/mi15081021_

Round 1

Reviewer 1 Report

Comments and Suggestions for Authors

Please find the attachment

Comments on the Quality of English Language

The English of this manuscript needs further improvement.

Author Response

Thank you very much for your comment. Please see to the attachment

Reviewer 2 Report

Comments and Suggestions for Authors

The paper presents some comprehensive and critical review on Research Status and Progress in  Grinding and Precision Machining of BK7 Glasses. However, the paper manuscript needs to undertake the following revisions:

(1) The paper is better titled as 'Comprehensive Review on Research Status and Progress in Precision Grinding and Machining of BK7 Glasses'.

(2) The Keywords should be better rearranged as 'BK7 glass; Hard and brittle materials; Precision grinding; Precision machining; Surface quality; Ultrasonic assisted machining'.

(3) Section 7 is better titled 'Conclusions and Future Research Trends'.

(4) The following very relevant papers and books in the topic area should be reviewed and included in References section as well:

Micro Cutting: Fundamentals and Applications, John Wiley & Sons, Chichester, October 2013.

- Investigation on Surface Morphology and Tribological Property Generated by Vibration Assisted Strengthening on Aviation Spherical Plain Bearings, Proceedings of the IMechE, Part C: Journal of Mechanical Engineering Science, 233(12), 2019, 4091-4101

Effect of self-developed graphene lubricant on tribological behaviour of silicon carbide/silicon nitride interface, Ceramics International, 45(8), 2019, 10211-10222.

Integrated modelling and analysis of micro-cutting mechanics with the precision surface generatIntegrated modelling and analysis of micro-cutting mechanics with the precision surface generation in abrasive flow machining, International Journal of Advanced Manufacturing Technology, 105, 2019, 4571-4583.

Comments on the Quality of English Language

(1) The paper is better titled as 'Comprehensive Review on Research Status and Progress in Precision Grinding and Machining of BK7 Glasses'.

(2) The Keywords should be better rearranged as 'BK7 glass; Hard and brittle materials; Precision grinding; Precision machining; Surface quality; Ultrasonic assisted machining'.

(3) Section 7 is better titled 'Conclusions and Future Research Trends'.

Author Response

Thank you very much for your comment. Please see to the attachment.
